# Towards artificial intelligence-based disease prediction algorithms that comprehensively leverage and continuously learn from real-world clinical tabular data systems

**Terrence J. Lee-St. John**[1]\*, **Oshin Kanwar**[1], **Emna Abidi**[1], **Wasim El Nekidy**[2], **Bartlomiej Piechowski-Jozwiak**[3,4]

1 Research Department, Cleveland Clinic Abu Dhabi, Abu Dhabi, United Arab Emirates, 2 Pharmacy Department, Cleveland Clinic Abu Dhabi, Abu Dhabi, United Arab Emirates, 3 Neurology Institute, Cleveland Clinic Abu Dhabi, Abu Dhabi, United Arab Emirates, 4 Neurology Department, Canberra Hospital, ACT, Australia

\* tjleestjohn@gmail.com

**Data Availability Statement:** As is standard with all real-world clinical patient data, the EHR data utilized during the PoC for this project is not

## Abstract

This manuscript presents a proof-of-concept for a generalizable strategy, the *full algorithm*, designed to estimate disease risk using real-world clinical tabular data systems, such as electronic health records (EHR) or claims databases. By integrating classic statistical methods and modern artificial intelligence techniques, this strategy automates the production of a disease prediction model that comprehensively reflects the dynamics contained within the underlying data system. Specifically, the full algorithm parses through every facet of the data (e.g., encounters, diagnoses, procedures, medications, labs, chief complaints, flow-sheets, vital signs, demographics, etc.), selects which factors to retain as predictor variables by evaluating the data empirically against statistical criteria, structures and formats the retained data into time-series, trains a neural network-based prediction model, then subsequently applies this model to current patients to generate risk estimates. A distinguishing feature of the proposed strategy is that it produces a self-adaptive prediction system, capable of evolving the prediction mechanism in response to changes within the data: as newly collected data expand/modify the dataset organically, the prediction mechanism automatically evolves to reflect these changes. Moreover, the full algorithm operates without the need for a-priori data curation and aims to harness all informative risk and protective factors within the real-world data. This stands in contrast to traditional approaches, which often rely on highly curated datasets and domain expertise to build static prediction models based solely on well-known risk factors. As a proof-of-concept, we codified the full algorithm and tasked it with estimating 12-month risk of initial stroke or myocardial infarction using our hospital's real-world EHR. A 66-month pseudo-prospective validation was conducted using records from 558,105 patients spanning April 2015 to September 2023, totalling 3,424,060 patient-months. Area under the receiver operating characteristic curve (AUROC) values ranged from .830 to .909, with an improving trend over time. Odds ratios describing model precision for patients 1–100 and 101–200 (when ranked by estimated risk) ranged from 15.3 to 48.1 and 7.2 to 45.0, respectively, with both groups showing improving trends over time.

publicly available due to HIPAA and local privacy laws. Deidentified extracts will be considered upon request from the Research Department at CCAD (Research@ClevelandClinicAbuDhabi.ae) but will require case-by-case approval from the CCAD Research Ethics Committee and Department of Health Abu Dhabi in alignment with institutional ethics policies and local privacy laws respectively.

**Funding:** The author(s) received no specific funding for this work.

**Competing interests:** The authors have declared that no competing interests exist.

Findings suggest the feasibility of developing high-performing disease risk calculators in the proposed manner.

## Author summary

The increasing prominence of disease risk calculators for preventive health is hindered by generalizability issues and narrow reliance on traditionally considered risk factors. We present a generalizable strategy/algorithm that automates the production of risk calculators that comprehensively leverage and continuously learn from real-world local tabular patient data systems, thereby ensuring that the resulting predictions reflect the contemporary health dynamics relevant to the local populations. Our local-focused approach contrasts with strategies that use increasingly extensive but less locally relevant data during model development. By applying our algorithm to a single hospital's electronic health record system, we demonstrate the feasibility of our approach, as well as the potential for it to produce high-performing risk estimation systems in a real-world setting. Though we present a particular specification, our approach is general and should be adapted to reflect the specifics of the local context and endpoint of interest. Additionally, because our strategy aims to utilize all available informative tabular data when calculating risk, as local tabular data systems evolve to include new sources of data (e.g., omics, health monitor, structured data extracted autonomously from physician notes or imaging studies), our strategy can be readily expanded to include these new data, potentially enhancing risk estimates.

## Introduction

Disease risk calculators serve as valuable tools in clinical practice, utilizing patient data to generate risk classifications (e.g., low vs high risk) or probabilities of disease onset within specific time frames. Widely used cardiovascular disorder (CVD) risk calculators such as QRISK3, Atherosclerotic Cardiovascular Disease Risk Score, and Framingham Risk Score exemplify this approach, offering 10-year risk estimates based on patient demographics and clinical data [1–3]. Integration of risk calculators into clinical practice for preventative purposes is increasingly common globally, with some even becoming standard of care, as is the case with QRISK3 which is now recommended by the UK Ministry of Health as part of standard CVD risk assessment [4,5].

However, despite widespread use, concerns around reductions in predictive accuracy (i.e., lack of generalizability) persists for real-world applications [6–14]. Even meticulously developed calculators, based on robust methodology and high-quality data, can encounter generalizability issues due to their reliance on correlated variables rather than direct physiological measurements of disease endpoints. Consequently, discrepancies may arise when applying these tools to populations or time periods distinct from those used during their original development.

Given this understanding, which is grounded in basic inferential statistical theory, practitioners must exercise caution when interpreting risk scores derived from calculators built on data from disparate contexts. In cases of significant dissimilarities, developing region-specific calculators trained on contemporary local data may be warranted to better capture current local health dynamics.

For instance, studies have shown that popular risk calculators designed using data from Western populations may not generalize well to the United Arab Emirates' (UAE) population, potentially leading to underestimation of CVD risk in younger Emirati adults [14–16]. Far from harmless, this has the potential to skew clinical assessment and delay the initiation of life-saving preventative treatments for UAE patients [14,17]. Similarly, a team at Oxford undertook an external validation of QRISK3 to examine how well it generalized to the UK Biobank cohort, calculating an area under the receiver operating curve (AUROC) of .710, down from an internal validation AUROC of .888 [1,18]. Because performance greatly decreased across two UK samples, this further raises questions about how reasonable it is to expect QRISK3 to generalize well to a more dissimilar population. However, even if one is willing to assume QRISK3 generalizes to one's local patients to the same degree as the UK Bio Bank, an AUROC of .710 indicates substantial room for improvement.

To enhance predictive performance, a recent systematic review of risk calculators suggests incorporating a larger set of predictor variables based on a more complete utilization of available clinical data, leveraging longitudinal/time-series data structures, and employing flexible non-parametric modelling techniques [19].

To ensure that health trends present in local populations are reflected in the prediction mechanism, our objective was to develop a generalizable disease risk estimation strategy/algorithm that can base prediction on information contained within a real-world clinical tabular data system, such as an electronic health record (EHR) or claims database. By combining classical statistical methods with modern artificial intelligence (AI) techniques, we created the overall strategy described in this manuscript, the *full algorithm*. Running a codification of the full algorithm on an underlying data system produces a unique predictive model that comprehensively reflects the dynamics contained within the data. The full algorithm autonomously parses through every facet of the data (e.g., encounters, diagnoses, procedures, medications, labs, chief complaints, flowsheets, vital signs, demographics, etc.). It then selects which factors to retain as predictor variables by evaluating the data empirically against statistical criteria, structures and formats the retained data into time-series for all patients, trains a neural network-based *final-stage prediction model*, and subsequently applies this model to current patients to generate risk estimates.

Due to its autonomous capabilities, scheduled rerunning of the full algorithm results in a self-adaptive system capable of evolving its prediction mechanism over time in response to changes in the underlying data. Such changes may be due to organic growth resulting from newly collected patient data, and can produce further alignment of the prediction mechanism with historical trends and/or shifts in the prediction mechanism reflecting newly emerging health dynamics. Thus, unlike traditional static predictive models derived from static historic datasets, the full algorithm automatically refines its risk estimates over time in a manner that reflects up-to-date clinical information contained within a live tabular data system.

Central to our strategy is the comprehensive utilization of the tabular data available within the local system to optimize prediction, including the leveraging of risk and protective factors that are both established and novel. By eschewing reliance on pre-defined sets of well-known predictors, the full algorithm can adapt to evolving clinical and data landscapes, ensuring robust and contextually relevant risk estimation.

Implied in this discussion is that our strategy attempts to leverage uncurated data systems. This is in contrast with traditional approaches, which rely on building models from highly curated datasets that contain minimal error. Though error free data are ideal, we postulate that our strategy has practical utility because producing curated datasets requires intensive manual a-priori data cleaning, and this reliance limits one from building an automated algorithm that can continuously learn from real-world systems. Additionally, limits on data curation

resources ensure that the resulting datasets will be low-dimensional relative to full EHR and claims databases. Thus, reliance on curated data may constrain the information available for prediction, resulting in suboptimal predictive performance.

We initially coded the full algorithm to run on our hospital's real-world EHR. However, because the general strategy outlined in this manuscript is not overly complex, it can readily be implemented on another data system by either (a) coding the strategy from inception locally using an open-source programming language (e.g., R), or (b) adapting our initial script to run on another data system with appropriate field mappings. In either instance, each run of the codified full algorithm will produce a unique prediction model reflecting the underlying dynamics contained within the data system at that point in time. Consequently, when implementing the full algorithm, performance should be expected to vary across both data systems and run timings.

In the subsequent sections we detail the development of the full algorithm when utilizing a month-level time-series to capture patient histories. The strategy can be applied to shorter (e.g., weekly) or longer (e.g., quarterly) intervals as well, and so generalizes to a diverse set of specifications. We discuss the rationale behind key development choices, though it is important to note that future algorithm specifications should reflect relevant local clinical and data/ IT considerations, and so may vary from what is presented here. Throughout we also illustrate implementation of the full algorithm in the context of a proof-of-concept (PoC) endpoint–initial stroke or myocardial infarction (MI) within 12 months. Validation of the full algorithm's performance and ability to learn over time is undertaken through a pseudo-prospective analysis spanning 66 months–here we show how predictive performance would have changed over this time range had the full algorithm been implemented at our hospital to predict the PoC endpoint monthly. Finally, we discuss broader implications and limitations.

It is essential to clarify that our objective was pragmatic rather than exploratory or scientific, and that our work represents an engineering endeavor. In this manuscript we simply aim to (1) describe a strategy we developed for creating self-adapting disease risk calculators that base prediction on, and continuously learn from, real-world clinical tabular data systems, (2) demonstrate the feasibility of implementing this strategy, and (3) illustrate how this strategy functions in a real-world use-case PoC.

## Ethics and information governance

The *Research Ethics Committee* (which serves as the IRB) at Cleveland Clinical Abu Dhabi (CCAD) approved all project protocols and granted a waiver of informed consent. All methods were performed in accordance with the relevant guidelines and regulations. Data utilized during this project never left CCAD's secured internal network. Full algorithm coding was entirely performed on the Research Department's dedicated analytics server, which has restricted access and restricted inter- and intranet connectivity aligned with CCAD's data privacy policies.

## Development methods

To validate the full algorithm, we employ a retrospective dataset while implementing an iterative, pseudo-prospective validation approach. Specifically, to generate risk estimates for patients accessing the hospital during month *T*, we execute a single run of the full algorithm using only the subset of the EHR corresponding to data collected from the hospital's inception through month *T-minus-1*. Subsequently, the resulting final-stage prediction model is applied to patients with hospital encounters in month *T*, constituting the *pseudo-prospective validation cohort*. The appropriate follow-up window is then observed, and model performance is

assessed. This process is then repeated for successive months (e.g., months *T-plus-1*, *T-plus-2*, *T-plus-3*) until reaching the most current pseudo-prospective validation cohort where the endpoint follow-up window is fully observable.

For the example PoC endpoint, this methodology was applied monthly from April 2017 to September 2022, yielding 66 distinct final-stage prediction models and 66 pseudo-prospective validation cohorts.

## Data, longitudinal structure, and outcome variable

The *full dataset* is sourced from the local EHR's backend relational database, necessitating the ability to query the EHR using SQL or a similar table extraction language. This dataset will encompass the entire available history up to the present day, inclusive of all structured clinical and demographic fields from every patient in the EHR who either (a) does not exhibit the endpoint or (b) exhibits the endpoint and has undergone at least one hospital encounter (e.g., an in- or outpatient visit, lab test, or medication order/fill) prior to the endpoint. For patients with the endpoint, this entails at least one hospital encounter occurs at least one month prior to the endpoint's month. Additionally, inclusion/exclusion criteria may be applied to further refine the dataset as necessary, with these criteria tailored to the project objectives, clinical context of the endpoint, and any pertinent legal, ethical, or contextual considerations.

In the case of the example PoC endpoint, the complete dataset originates from CCAD's EHR relational database (Epic Caboodle), spanning from April 2015 (the first full month of CCAD operations) through September 2023. It encompasses all structured clinical and demographic fields from every CCAD patient aged 18 or older who had at least one hospital encounter and had no prior history of stroke/MI before their initial CCAD encounter, which itself was not associated with a stroke/MI. Notably, due to minimal inclusion/exclusion criteria, this PoC represents an application of the full algorithm aimed at creating a universal screening system for CCAD's endpoint-naïve adult population.

Patients' longitudinal clinical histories are condensed into patient-month-level time-series, where patient-months are established solely for calendar months in which a patient has at least one hospital encounter. A binary outcome variable at the patient-month level indicates the presence of the endpoint (example PoC: ICD-10 codes G45, I63-I64, I20-I25) within the relevant follow-up period (example PoC: 12 months). This outcome is only coded for patient-months where the full follow-up period is observable. For patients who do not experience the endpoint but pass away before the follow-up period's completion, the outcome is coded as not occurring–this decision is subject to modification based on the desired prediction goals, however. Time-series for patients who experience the endpoint are censored after the final patient-month preceding the endpoint month.

Regarding the example PoC, the resultant data splits and their utilization for the pseudo-prospective validation cohort from month *T* are outlined as follows:

- The final-stage prediction model is trained/tested on the *analytic dataset* comprising data spanning April 2015 through month *T-minus-13*, with months *T-minus-12* through *T-minus-1* utilized solely for observing/coding of the outcome.

  ○ For training purposes, the analytic dataset is randomly partitioned into *training* and *test* datasets using a 90:10 ratio at the patient-level.

  ○ Patients with encounters during month *T* serve as the pseudo-prospective validation cohort, with months *T-plus-1* through *T-plus-12* utilized solely for observing/coding the outcome for these validation patients.

**Table 1. PoC Data Splits.**

| | Full Dataset | | | | | | | | | | | |
|---|---|---|---|---|---|---|---|---|---|---|---|---|
| | Apr-15 | | | | | | | | | | | Sep-23 |
| | Analytic Dataset (split into Training and Test Datasets) | | | 12-Month Follow-Up for Analytic Dataset | | | Pseudo-Prospective Validation Cohort | 12-Month Follow-Up for Pseudo-Validation Dataset | | | Unused Dataset | | |
| Vignette 1 | Apr-15 | ... | Mar-16 | Apr-16 | ... | Mar-17 | Apr-17 | May-17 | ... | Apr-18 | May-18 | ... | Sep-23 |
| Vignette 2 | Apr-15 | ... | Dec-19 | Jan-20 | ... | Dec-20 | Jan-20 | Feb-20 | ... | Jan-21 | Feb-21 | ... | Sep-23 |
| Vignette 3 | Apr-15 | ... | Aug-21 | Sep-21 | ... | Aug-22 | Sep-22 | Oct-22 | ... | Sep-23 | - | ... | - |

○ Months *T-plus-13* though the end of the full dataset (i.e., September 2023) remain unused.

Table 1 illustrates a visual representation of these data splits for three vignettes, each derived from a distinct pseudo-prospective validation cohort in the example PoC.

For the example PoC, the full dataset comprises records from 558,105 patients spanning 3,424,060 patient-months. Patient Ns for the 66 pseudo-prospective validation cohorts and their corresponding analytic datasets are detailed in Table 2 in the Results section below.

## General representation of predictor data

The potential predictor space encompasses all diagnoses (ICD-10 codes), procedures (CPT codes), chief complaints, medications, labs, flowsheets, vital signs, encounter characteristics, and demographics present in the analytical dataset.

Binary variables alone suffice to denote the presence or absence of categorical information in a patient record (e.g., ICD-10 code *X* present = 1, *else* 0).

Conversely, capturing a numeric measurement where some patients lack the measurement in question requires a paired combination of a binary and a numeric variable. The binary variable indicates the presence or absence of the numeric measurement, while the numeric variable stores either the precise numeric value for patients with the measurement or a constant numeric value for those without it. This paired coding scheme (a) eliminates case-wise deletion due to the absence of numeric measurements in some patients (e.g., those who do not received lab *X*), and (b) fully partials out the influence of patients without the measurement from contributing to the coefficients/weights associated with the numeric variable. Thus, though a constant numeric value is inserted for patients without the measurement, this type of imputation does not modify the underlying patient information in a manner that affects risk estimates.

It is crucial to acknowledge that, while implicit, no missing data exist given these data representations. Data errors exist if the presence or absence of a code does not accurately reflect clinical reality; however, such discrepancies are not manifested as missing values in the predictor variables. Rather, the representations described here faithfully mirror the actual real-world data, obviating the need for further imputation of missing values (beyond the numeric variable coding scheme described above) to prevent case-wise deletion.

## Considerations related to high-dimensional predictor-space

Due to the detailed nature of EHR or claims systems, the patient-level analytical dataset may potentially encompass tens of thousands of unique predictors. For instance, in the PoC, the potential number of predictors exceeds 32k, including approximately 15.3k ICD-10 names, 9.3k CPT names, 1k chief complaints, 642 medication classes, 2.6k labs, 3.8k flowsheet categories, and 25 demographic variables. While solely utilizing binary variables to represent this high-dimensional predictor space is a more practical choice due to their lower resource

demands compared to numeric data, doing so inevitably entails information loss which could detrimentally affect prediction accuracy.

At this stage, we theorized that due to the nature of patient clinical journeys, contemporary clinical variables are interrelated and also reflect previous clinical encounters within a patient's journey. Hence, we anticipated cross-sectional and longitudinal correlations among variables within a patient's record. If reasonable, this expectation suggests that representative patient histories can be approximated without including numeric values because discarded numeric information will be at least partially represented in both contemporary and longitudinally correlated binary variables. For example, imagine a patient attends a primary care visit, is ordered a complete blood panel and HBA1C test, receives a HBA1C numeric lab result that is above the threshold to diagnoses diabetes, is subsequently diagnosed with diabetes, then is scheduled an endocrinology appointment, and is prescribed a blood glucose medication. If we then choose to exclude all numeric lab results, the series of events still clearly reflects the patient's journey and even reflects the excluded HBA1C value (albeit imprecisely). Given these considerations, we opted for a primarily-binary coding scheme to ensure that implementing the full algorithm was pragmatic given our computer resource capacities. Specifically, we chose to specify binary variables to denote the presence of all categorical predictors (e.g., procedure $X$ present = 1, *else* 0) and the majority of numeric measurements (e.g., lab $X$ measurement present = 1, *else* 0).

Despite this choice, dimensionality reduction of the predictor space may still be necessary due to practical limits related to computer resources. Additionally, many variables may not contribute significantly to prediction and therefore represent *noise* in the predictor space if retained. Retaining excessive noise threatens the model's generalizability, as the true prediction signal may become obscured during training. For example, if a binary predictor (BP) spuriously indicates a 0% risk association with the outcome due to random sampling error, the trained model may incorrectly assign a 0% risk estimate to patients with this predictor regardless of the other values in their predictor space, effectively incorrectly eliminating them from being considered high risk on the basis of a single BP.

Hence, our aim is to reduce the binary predictor space in a way that preserves sufficient relevant information to capture the true prediction signal while eliminating enough noise to enable the identification of this signal. To achieve this automatically, we employed classic statistical hypothesis-testing and sampling distributional logic to apply two *predictor retention criteria* (PRC) at the patient level to each BP. Those meeting both criteria are automatically retained and integrated into the final-stage prediction model.

Finally, we acknowledge that while we have primarily discussed challenges associated with the high-dimensional predictor space, we propose that high-dimensionality in this context could also serve as a potential asset for prediction. Specifically, considering that real-world clinical tabular data systems inherently exhibit high correlation and high dimensionality, it is plausible that the disease prediction signal can navigate through a predictor-space containing errors or noise because there are multiple complementary pathways available for the prediction signal to traverse. Essentially, we operate under the assumption that high-dimensional and correlated predictor spaces can yield a predictive mechanism that remains robust to errors in individual predictive pathways. Although not directly investigated in this project, empirical findings from our PoC, as discussed below, suggest the plausibility of this hypothesis.

## Retained numeric predictors

We opted to consistently capture and retain a small set of numeric predictors describing age, vital signs (e.g., BMI, blood pressure), and longitudinal time-series descriptors (e.g., time

between consecutive patient-months, numbers of encounters within a patient month) because these numeric variables are common and sourced from low-dimensional tables within our EHR's relational database, thus imposing minimal additional computational resource requirements.

### Automated binary Predictor Retention Criterion #1 (PRC-1)

This criterion was formulated to ensure that BPs were retained only if their observed Phi correlations with the outcome in the analytic dataset were unlikely to be due to random sampling error and were thus likely to contain information that can be utilized to predict the outcome beyond the analytic dataset to unseen data. Specifically, for each BP we conducted Fisher's Exact Test to compute the two-tailed p-value for the associated 2-by-2 contingency table (BP-by-outcome). A predictor meets PRC-1 if the two-tailed $p \leq .05$. It is important to note that with this specification, 2.5% of the sampling distribution within each tail meets PRC-1: one tail pertains to *protective factors* (BPs that decrease risk of the endpoint), and the other to *risk factors* (BPs that increase risk of the endpoint).

The exact alpha/p-value criteria for retention here can be considered a hyper-parameter in the full algorithm–adjustment will result in a larger or smaller retained BP-space with varying degrees of predictive information and noise. Additionally, asymmetric alpha criteria can be specified if one wants to prioritize retention of risk factors over that of protective factors (or vice versa).

As a direct result of applying Fisher's Exact Test, for a protective factor to be retained it must have been present in at least $N_{ProtX\_min} = ceiling\left(\frac{\log\left(\frac{.05}{2}\right)}{\log(1-rate)}\right)$ patients in the analytic dataset, where *rate* is the patient-level incidence rate of the outcome in the analytic dataset, and the *ceiling*() function rounds up to the nearest integer. $N_{ProtX\_min}$ represents the minimum patient N for *complete* protective factors, those where 0% of patients who have the factor also have the outcome. For protective factors that do not imply complete protection, the minimum patient N is a function of endpoint rates for those with and without the factor, as determined by Fisher's Exact Test, but will always be larger than $N_{ProtX\_min}$.

In the example PoC across all analytic datasets, patient-level incidence rates of the endpoint range from 2.28% to 2.78%, thus, $N_{ProtX\_min}$ varies from 160 to 131 for complete protective factors.

We opted for Fisher's Exact Test over the more commonly known Chi-Squared Test because the resulting p-values are valid regardless of sample size, and therefore apply appropriately to both common and rare BPs.

Additionally, we acknowledge that due to the current specification, the familywise error rate for the set of BPs retained by PRC-1 is expected to exceed .05. However, we do not consider this an issue because unlike traditional inferential statistical exercises, our goal is not to create a parsimonious parametric model for the sake of interpreting individual coefficients in a causal or pseudo-causal manner. Instead, PRC-1 primarily aims to reduce the noise in the predictor-space to a level that allows the final-stage prediction model to roughly uncover the prediction signal. As discussed later, to mitigate the negative effect of the retained noise, the final-stage prediction model employs a neural network architecture and estimation strategy that diminishes the influence of non-generalizable predictor data.

### Automated binary Predictor Retention Criterion #2 (PRC-2)

As a result of applying Fisher's Exact Test, PRC-1 may retain a BP with a sample size $< N_{ProtX\_min}$ if the BP is a risk factor and its Phi correlation with the endpoint is substantial. In extreme

cases, this has the potential to compromise the generalizability of the training dataset to the test dataset during the model training process. For instance, when the observed correlation of a risk factor in the analytic dataset is unlikely to be observed in a randomly selected test dataset simply due to its small associated sample size, this discrepancy between the training and test datasets can impede model convergence during training and may result in a model that inadequately captures the prediction signal.

Therefore, PRC-2, which exclusively applies to risk factors, aims to ensure that risk factors are retained only when their expected sample sizes in a randomly selected test dataset are sufficiently large to reasonably expect that the observed effects in the analytic dataset will partially generalize to the test dataset empirically. Specifically, for risk factor $X$, define $N_{X=1}$ as the number of patients in the analytic dataset with risk factor $X$, $O_{X=1}$ as the number of patients in the analytic dataset with both risk factor $X$ and the outcome, $O_{X=1test}$ as the test dataset equivalent of $O_{X=1}$, and $N_{X=1test} = .10*N_{X=1}$ as the test dataset equivalent of $N_{X=1}$. Then, given that $\frac{O_{X=1}}{N_{X=1}}$ is a true parameter value for the analytic dataset by definition, and that $N_{X=1test}$ is a random sample from the analytic dataset, we calculate the exact probability that $\frac{O_{X=1test}}{N_{X=1test}} > 0$ using the cumulative binomial distribution. Risk factor $X$ meets PRC-2 when this probability is $\geq$95%.

Like the alpha/p-value criterion from PRC-1, the exact probability criterion for retention during PRC-2 can also be considered a hyper-parameter in the full algorithm–adjusting this value will result in a larger or smaller retained BP space with varying degrees of expected discordance between training and test datasets.

When combined, PRC-1 and PRC-2 may still permit the retention of risk factors with very small sample sizes in the analytic dataset. However, this only occurs when their effect sizes are sufficiently large to reasonably expect partial generalization to the test dataset. The exact minimum sample size for a risk factor is determined by the statistical methodology employed here and thus depends on the magnitude of the effect size found in the analytic dataset. Conceptually, risk factors with larger effect sizes will have smaller minimum N requirements compared to those with smaller effect sizes.

## Visualization of PRC-1 and PRC-2

A visualization of the automated BP selection process for clinical risk factors in the PoC is depicted in Fig 1. Here, we present the joint distribution of PRC-1 values (x-axis) and PRC-2 values (y-axis) calculated for each potential binary clinical risk factor for the September 2022 pseudo-prospective validation cohort. Points are color-coded to represent different types of clinical data (e.g., Diagnoses, Labs). Reference lines for cut points 5% and 95% are placed on the axis corresponding to PRC-1 and PRC-2 retention criteria respectively. Points falling at or to the left of the vertical reference line meet PRC-1 criteria, while those falling at or above the horizontal reference line meet PRC-2 criteria. Risk factors located within the upper-left region of the graph, delineated by the two reference lines, satisfy both criteria and are retained during automated predictor selection.

Overall, for the example PoC our predictor selection strategy reduced the predictor-space from over 32k to between 966 and 4,150 depending on the pseudo-prospective validation cohort. The exact number of predictors associated with each cohort is detailed in Table 2 in the Results section. Additionally, the precise list of retained predictors for the final pseudo-prospective validation cohort (September 2022) is presented in S1 Appendix. An examination of this list reveals the detailed nature of the EHR data incorporated into the final-stage prediction model.

## Predictor encoding schema–Capturing historic and contemporary data simultaneously

In each patient's time-series, retained diagnoses are marked as present from the initially occurring patient-month onwards (e.g., if diagnosis *X* occurred during or prior to this month, then diagnosis *X* = 1, *else* 0). This *past and present coding scheme* was exclusively employed for diagnoses due to the hospital's workflow, where the same diagnosis is frequently entered across subsequent encounters. Hence, except for initially occurring codes, observed diagnoses often reflect ongoing conditions, especially chronic ones.

It is worth noting that because this strategy disregards accurate information on recurring acute diagnoses (e.g., infections), an alternative coding scheme for diagnoses might be preferable if the endpoint of interest is sensitive to recurring diagnostic data. Such decisions should be made on a case-by-case basis guided by clinical understanding, and can be empirically evaluated by testing alternative specifications using the same pseudo-prospective validation methods we employ.

All other retained BPs representing clinical data are encoded into the longitudinal structure in two ways. The first employs a *historical coding scheme* where predictor values only describe historical presence (e.g., if medication class *X* was administered prior to this month, then medication class *X* = 1, *else* 0). The second utilizes a *contemporaneous coding scheme* where predictor values describe the current patient-month status only (e.g., if medication class *X* was administered during this month, then medication class *X* = 1, *else* 0).

For categorical demographic variables (e.g., current smoker, past smoker, never smoker), dummy series are initially coded with an added binary indicator for missing data (e.g., smoking status missing = 1, *else* 0). Time-varying demographics are then encoded into the time-series structure using the contemporaneous coding scheme, while time-invariant demographics are coded as constants across all rows within a patient's time-series.

For each retained numeric measurement except age, we first specify a binary variable indicating the presence or absence of the numeric measurement, alongside a second numeric variable indicating the precise numeric value for those with the measurement and a constant value for those without. In cases where a patient received multiple measurements within a single patient-month, we utilize the last chronological value. However, for age, only a single numeric variable is specified because it is universally captured in our hospital's EHR and so an accompanying binary indicator would be constant/redundant.

For all numeric measurements except age (birth date is accurately captured in our EHR), extreme low and high values are trimmed to the 0.1 and 99.9 percentiles respectively based on the analytic dataset. Subsequently, age and trimmed variables are standardized using the mean and standard deviation calculated from the analytic dataset. In instances where a numeric measurement is absent, we then enter 0 as the constant numeric value, effectively positioning these points at the mean of the previously standardized variables.

All resulting variables except age are coded into the time-series structure using both the historical and contemporaneous schema. For numeric variables, the maximum historic value to date is utilized as the historic encodings. For age, only the contemporaneous value is utilized.

When applied to the example PoC, these predictor encoding schemas expand the predictor-space from 966 to 4,150 retained predictors to 1,778 to 7,481 predictor encodings within the patient-month time-series structure.

## Neural Network (NN) structure

When selecting a NN structure, we considered the inherent errors in real-world data and the likelihood of retaining noise in the predictor-space through automated prediction selection.

Hence, while the final-stage prediction model should roughly reflect the true prediction signal, it is also crucial for the model not to precisely replicate the exact patterns in the analytic dataset.

In general, opting for a simpler structure limits the NN's capacity to capture specific patterns from the analytic data. Additionally, incorporating L2 kernel regularization (analogous to ridge regression) shrinks/diminishes the influence of predictor encodings. This helps manage collinearity in the predictor-space, aids model estimation with an overspecified predictor-space, and can bolster generalizability to unseen data (provided the prediction signal is not significantly dampened by the shrinkage) [20].

Considering these factors, we opted for a simple shallow feed-forward NN in the final-stage prediction model: it features an input layer comprising predictor encodings, a single densely-connected hidden layer with 32 nodes each employing sigmoid activations and L2 = 0.0001 kernel regularizations, and a final densely-connected output layer with 1 node using a sigmoid activation. The estimates generated by the final output layer are on the logit scale, which are then transformed into risk values (i.e., probabilities of the outcome) using the standard logit link function.

It is important to note that the number of nodes in the hidden layer and the L2 norm value can be considered hyperparameters in the full algorithm. Adjustments to these parameters will impact the NN's ability to capture the precise patterns in the analytic dataset and its generalizability to unseen data.

## NN training

After selecting and encoding the predictor-space using the described strategy, we split the analytic dataset into training and test datasets with a 90:10 probability split at the patient level. Initially, training is conducted with a patient-month batch size of $2^7$, implementing early stopping with a patience of 2 based on test dataset loss. Subsequently, the patient-month batch size is increased to $2^{14}$, and early stopping (with patience = 4) is again applied during this phase based on test dataset loss. Throughout both steps, the Adamax gradient descent algorithm is employed for NN training, and binary cross-entropy serves as the loss function. Early stopping is utilized to prioritize generalizability, addressing potential data errors and noise in the predictor-space.

It is of practical importance to note that because our predictor encoding schema are designed to capture both historic and contemporary data in a patient's time-series simultaneously, this eliminates the need for computationally demanding NN architectures such as transformers or memory constructs that pull historical data influence forward across time-series rows. Consequently, we can utilize the simple NN structure described above, minimizing the computer resources required for NN training. In the example PoC, training times for final-stage prediction models increase from month to month as additional data are integrated into successive runs of the full algorithm. However, even in the largest run (September 2022 validation cohort), the NN training process was completed in under one hour using a single PCIe form GPU.

## Summary of full Algorithm

A visual summary of a single run of the full algorithm is presented in Fig 2. We split the full algorithm into three sections.

1. Pre-Process Historic Data

○ Specify

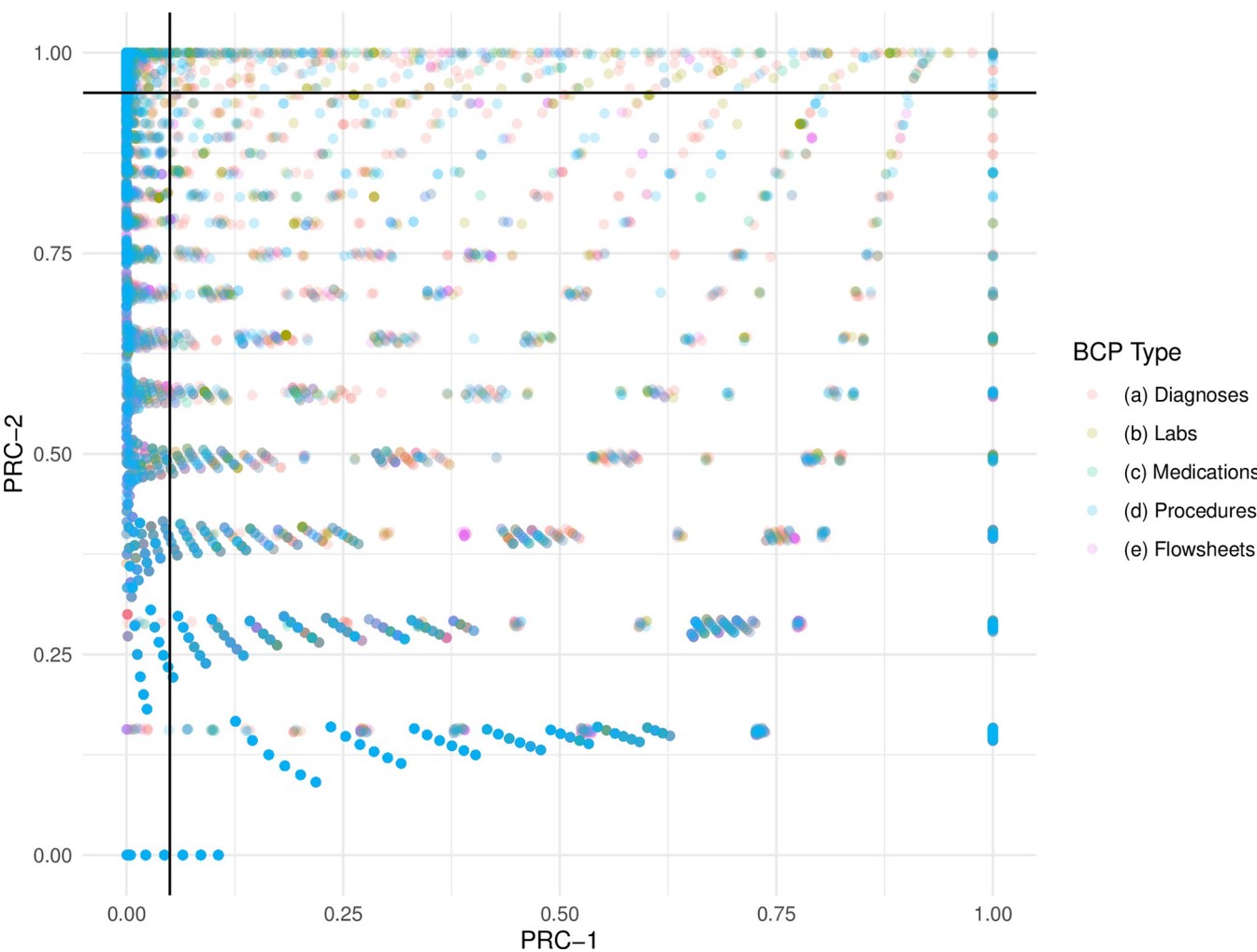

**Fig 1. Predictor Retention Criteria for Binary Risk Factors. Pseudo-Prospective Validation Cohort = September 2022.**

- ■ Time-Series Structure

- ■ Endpoint

- ■ Population of Interest

- ○ Automated Predictor Selection

- ○ Data Structuring and Formatting

2. Train Final Stage Prediction Model

3. Apply Model

This visualization shows the simplicity of our strategy, which suggests the feasibility with which it can be implemented locally. All relevant historical patient data are examined, then informative predictor variables are identified and retained from the real-world data system and directly encoded/formatted into the chosen time-series structure. These encoded data are then used to model disease risk using a feed-forward neural network, where a simple NN structure coupled with L2 kernel regularization are utilized to enhance generalization. The

resulting model is then applied to current patients from the same real-world data system to estimate their risk of the specified disease endpoint.

## Operationalizing risk estimates–Decision making strategies

In scenarios where the prediction model will be used to determine which patients to screen manually, two strategies are available: (1) employing a *threshold-based classification* to label patients as low or high risk, followed by manual screening of all high-risk patients, and (2) implementing *rank-ordered screening*, where patients are ranked by their estimated risk values, with manual screening beginning with the highest risk patient, proceeding downward. The rank-ordered strategy is feasible when processing data in large patient batches, as in the example PoC.

For the threshold-based strategy, properties of different thresholds can be examined using historical data, then set according to project needs. Conversely, when resources for manual screening are limited, the rank-ordered strategy aligns well with operational constraints by directing resources to patients most in need. Furthermore, since model performance is expected to be strongest for patients with higher risk scores, the rank-ordered approach tends to maximize model performance for a given number of screened patients.

## Hardware and software

A single PC workstation with dual CPUs, dual PCIe form GPUs, and 767 GB of system RAM was used during development. The full algorithm was coded entirely in the open-source language R 4.0.2 [21]. Heavily utilized R packages include data.table 1.12.8, keras 2.3.0.0, tensorflow 2.2.0, fastDummies 1.6.1, lubridate 1.7.9, and RODBC 1.3–17 [22–27].

## Results

Table 2 provides the 66 pseudo-prospective validation cohort AUROC values, area under the precision recall curve (AUPRC) values, positive predictive values (PPV or precision) and their associated odds ratios (OR) for patient groups 1–100 and 101–200 when ranked by highest estimated risk, as well as the total number of retained predictors, predictor encodings, and patients for all data splits (and the respective %'s of patients with the outcome). For example, for the September 2022 pseudo-prospective validation cohort, the analytic dataset came from 354,928 patients, and the model incorporated 4,150 retained predictors formatted into 7,481 predictor encodings. Validation was performed using data from 45,293 patients and achieved an AUROC of .874, an AUPRC of .108, and PPV(OR) for patients 1–100 and 101–200 of 27 (28.9) and 22(22.0) respectively.

Fig 3 provides a visual representation of the AUROC data and includes the linear trend line (fit using OLS regression) for reference. The AUROC range is .830 to .909. Within this range, examination of the trend reveals an upward/improving trajectory over time.

Fig 4 presents a visual representation of the AUPRC data, including a linear trend line (fit using OLS regression) for reference. The AUPRC range is .087 to .140. Within this range, examination of the trend reveals a downward/worsening trajectory over time.

Fig 5 presents a visual representation of the PPV odds ratios for patient groupings 1–100 and 101–200, including linear trend lines (fit using OLS regression) for reference. OR value ranges for patients 1–100 and 101–200 are 15.3 to 48.1 and 7.2 to 45.0 respectively. As expected, OR values for patients 1–100 tend to be higher than those for patients 101–200. Examination of the trends reveal similar upward/improving trajectories over time for both groups of patients.

**Table 2. Predicting 12-Month Initial Stroke/MI: Model Performance Across Successive Monthly Refreshes.**

| Pseudo-Pro. Val. Cohort Month | Analytic Dataset Patients | | Retained Predictors | Predictor Encodings | Pseudo-Pro. Val. Cohort Patients | | | | | | | |
|---|---|---|---|---|---|---|---|---|---|---|---|---|
| | N | % w/ Outcome | N | N | N | % w/ Outcome | AUROC | AUPRC | Patients Ranked by Highest Estimated Risk | | | |
| | | | | | | | | | #1–100 | | #101–200 | |
| | | | | | | | | | PPV | OR | PPV | OR |
| Sep-22 | 354,928 | 2.42 | 4,150 | 7,481 | 45,293 | 1.37 | .874 | .108 | 27 | 28.9 | 22 | 22.0 |
| Aug-22 | 338,204 | 2.50 | 4,110 | 7,410 | 44,084 | 1.36 | .875 | .113 | 28 | 30.8 | 22 | 22.3 |
| Jul-22 | 326,700 | 2.55 | 4,055 | 7,307 | 40,800 | 1.36 | .874 | .118 | 30 | 34.3 | 22 | 22.6 |
| Jun-22 | 313,989 | 2.61 | 4,014 | 7,238 | 46,355 | 1.39 | .868 | .106 | 26 | 27.0 | 21 | 20.4 |
| May-22 | 303,234 | 2.66 | 3,961 | 7,141 | 60,333 | 1.10 | .873 | .092 | 25 | 32.2 | 19 | 22.6 |
| Apr-22 | 296,804 | 2.68 | 3,903 | 7,039 | 38,680 | 1.29 | .881 | .117 | 28 | 32.7 | 16 | 16.0 |
| Mar-22 | 292,917 | 2.66 | 3,835 | 6,923 | 47,101 | 1.20 | .873 | .089 | 19 | 20.7 | 19 | 20.7 |
| Feb-22 | 289,597 | 2.65 | 3,811 | 6,886 | 47,516 | 1.17 | .873 | .088 | 17 | 18.5 | 18 | 19.8 |
| Jan-22 | 285,432 | 2.64 | 3,748 | 6,765 | 58,847 | 1.00 | .889 | .093 | 19 | 25.0 | 22 | 30.1 |
| Dec-21 | 281,613 | 2.63 | 3,703 | 6,684 | 51,856 | 1.05 | .882 | .100 | 25 | 34.3 | 21 | 27.4 |
| Nov-21 | 275,763 | 2.65 | 3,647 | 6,586 | 73,422 | 0.75 | .909 | .093 | 20 | 35.1 | 15 | 24.8 |
| Oct-21 | 265,778 | 2.70 | 3,572 | 6,460 | 79,577 | 0.70 | .906 | .092 | 24 | 48.1 | 15 | 26.9 |
| Sep-21 | 258,775 | 2.72 | 3,488 | 6,319 | 72,940 | 0.76 | .901 | .093 | 20 | 35.6 | 24 | 45.0 |
| Aug-21 | 255,526 | 2.71 | 3,398 | 6,166 | 64,934 | 0.82 | .900 | .098 | 23 | 39.0 | 17 | 26.8 |
| Jul-21 | 252,739 | 2.69 | 3,388 | 6,147 | 52,375 | 0.91 | .886 | .104 | 24 | 37.1 | 10 | 13.1 |
| Jun-21 | 249,579 | 2.69 | 3,255 | 5,891 | 57,496 | 0.89 | .893 | .106 | 27 | 45.2 | 16 | 23.3 |
| May-21 | 246,789 | 2.68 | 3,195 | 5,779 | 50,266 | 1.02 | .890 | .118 | 29 | 43.4 | 15 | 18.7 |
| Apr-21 | 236,770 | 2.76 | 3,143 | 5,681 | 39,498 | 1.16 | .882 | .111 | 22 | 26.2 | 17 | 19.1 |
| Mar-21 | 232,362 | 2.76 | 3,103 | 5,613 | 39,857 | 1.34 | .862 | .108 | 24 | 25.3 | 20 | 20.0 |
| Feb-21 | 228,770 | 2.75 | 3,071 | 5,555 | 38,064 | 1.33 | .862 | .113 | 27 | 30.2 | 20 | 20.4 |
| Jan-21 | 224,715 | 2.74 | 3,036 | 5,494 | 37,146 | 1.39 | .862 | .109 | 29 | 31.9 | 17 | 16.0 |
| Dec-20 | 220,143 | 2.75 | 2,996 | 5,428 | 32,589 | 1.51 | .858 | .117 | 28 | 28.0 | 18 | 15.8 |
| Nov-20 | 215,499 | 2.76 | 2,948 | 5,344 | 37,034 | 1.35 | .868 | .122 | 32 | 38.5 | 22 | 23.1 |
| Oct-20 | 209,599 | 2.77 | 2,898 | 5,253 | 43,356 | 1.22 | .875 | .109 | 28 | 34.7 | 22 | 25.2 |
| Sep-20 | 205,267 | 2.78 | 2,837 | 5,143 | 41,375 | 1.29 | .856 | .087 | 19 | 19.3 | 15 | 14.5 |
| Aug-20 | 202,352 | 2.78 | 2,775 | 5,026 | 31,603 | 1.53 | .843 | .107 | 27 | 26.0 | 13 | 10.5 |
| Jul-20 | 198,287 | 2.78 | 2,732 | 4,953 | 30,268 | 1.55 | .840 | .099 | 20 | 17.2 | 15 | 12.2 |
| Jun-20 | 194,914 | 2.76 | 2,688 | 4,873 | 32,491 | 1.44 | .836 | .088 | 22 | 20.8 | 12 | 10.0 |
| May-20 | 191,726 | 2.76 | 2,674 | 4,848 | 24,596 | 1.61 | .845 | .109 | 23 | 20.5 | 18 | 15.0 |
| Apr-20 | 188,089 | 2.75 | 2,643 | 4,787 | 34,238 | 1.16 | .874 | .097 | 19 | 22.0 | 17 | 19.2 |
| Mar-20 | 184,370 | 2.74 | 2,589 | 4,692 | 32,654 | 1.29 | .834 | .093 | 23 | 24.8 | 8 | 7.2 |
| Feb-20 | 180,835 | 2.73 | 2,546 | 4,618 | 30,409 | 1.39 | .839 | .098 | 23 | 23.1 | 13 | 11.6 |
| Jan-20 | 176,459 | 2.71 | 2,532 | 4,606 | 31,124 | 1.29 | .839 | .110 | 29 | 34.9 | 13 | 12.8 |
| Dec-19 | 171,278 | 2.69 | 2,490 | 4,528 | 31,245 | 1.37 | .848 | .107 | 26 | 27.8 | 12 | 10.8 |
| Nov-19 | 167,512 | 2.69 | 2,459 | 4,466 | 31,599 | 1.41 | .849 | .129 | 30 | 33.6 | 18 | 17.2 |
| Oct-19 | 162,827 | 2.70 | 2,409 | 4,377 | 36,835 | 1.30 | .865 | .119 | 27 | 31.2 | 21 | 22.5 |
| Sep-19 | 158,646 | 2.68 | 2,373 | 4,314 | 33,247 | 1.38 | .861 | .111 | 24 | 25.0 | 21 | 21.0 |
| Aug-19 | 155,424 | 2.69 | 2,366 | 4,307 | 26,678 | 1.58 | .859 | .117 | 29 | 28.7 | 19 | 16.5 |
| Jul-19 | 151,438 | 2.67 | 2,348 | 4,279 | 31,422 | 1.48 | .857 | .110 | 29 | 29.9 | 12 | 10.0 |
| Jun-19 | 148,698 | 2.64 | 2,310 | 4,211 | 28,646 | 1.64 | .856 | .117 | 25 | 22.1 | 18 | 14.5 |
| May-19 | 145,555 | 2.63 | 2,259 | 4,122 | 26,499 | 1.54 | .866 | .120 | 29 | 28.9 | 11 | 8.8 |
| Apr-19 | 141,662 | 2.62 | 2,231 | 4,068 | 30,508 | 1.61 | .854 | .112 | 23 | 20.2 | 22 | 19.1 |

*(Continued)*

**Table 2.** (Continued)

| Pseudo-Pro. Val. Cohort Month | Analytic Dataset Patients | | Retained Predictors | Predictor Encodings | Pseudo-Pro. Val. Cohort Patients | | | | | | | |
|---|---|---|---|---|---|---|---|---|---|---|---|---|
| | N | % w/ Outcome | N | N | N | % w/ Outcome | AUROC | AUPRC | Patients Ranked by Highest Estimated Risk | | | |
| | | | | | | | | | #1–100 | | #101–200 | |
| | | | | | | | | | PPV | OR | PPV | OR |
| Mar-19 | 137,649 | 2.60 | 2,174 | 3,973 | 30,243 | 1.61 | .860 | .115 | 26 | 23.6 | 18 | 14.7 |
| Feb-19 | 133,996 | 2.57 | 2,110 | 3,866 | 27,915 | 1.77 | .861 | .136 | 32 | 29.2 | 19 | 14.5 |
| Jan-19 | 129,169 | 2.55 | 2,042 | 3,745 | 29,748 | 1.68 | .853 | .116 | 26 | 22.5 | 18 | 14.1 |
| Dec-18 | 123,527 | 2.55 | 2,005 | 3,679 | 29,460 | 1.68 | .843 | .105 | 23 | 19.0 | 14 | 10.3 |
| Nov-18 | 119,981 | 2.53 | 1,945 | 3,573 | 27,715 | 1.54 | .850 | .096 | 18 | 15.3 | 17 | 14.3 |
| Oct-18 | 114,936 | 2.51 | 1,903 | 3,495 | 33,002 | 1.44 | .856 | .099 | 19 | 17.3 | 14 | 12.0 |
| Sep-18 | 111,224 | 2.49 | 1,847 | 3,396 | 28,886 | 1.64 | .855 | .114 | 25 | 22.0 | 18 | 14.5 |
| Aug-18 | 107,807 | 2.47 | 1,797 | 3,304 | 23,936 | 1.68 | .852 | .131 | 26 | 23.1 | 17 | 13.5 |
| Jul-18 | 104,169 | 2.46 | 1,743 | 3,207 | 26,878 | 1.78 | .853 | .117 | 23 | 18.2 | 21 | 16.2 |
| Jun-18 | 101,834 | 2.46 | 1,699 | 3,129 | 22,236 | 1.97 | .849 | .140 | 32 | 26.3 | 14 | 9.1 |
| May-18 | 98,000 | 2.44 | 1,651 | 3,047 | 24,417 | 1.70 | .845 | .133 | 28 | 25.2 | 15 | 11.4 |
| Apr-18 | 94,415 | 2.41 | 1,570 | 2,904 | 26,215 | 1.64 | .830 | .109 | 25 | 22.0 | 15 | 11.7 |
| Mar-18 | 90,876 | 2.37 | 1,500 | 2,773 | 25,291 | 1.76 | .844 | .134 | 33 | 30.6 | 13 | 9.3 |
| Feb-18 | 87,261 | 2.35 | 1,438 | 2,658 | 24,471 | 1.86 | .837 | .124 | 28 | 22.8 | 15 | 10.3 |
| Jan-18 | 81,927 | 2.34 | 1,366 | 2,520 | 26,532 | 1.73 | .846 | .109 | 22 | 17.8 | 23 | 18.9 |
| Dec-17 | 75,103 | 2.35 | 1,332 | 2,462 | 25,904 | 1.71 | .838 | .119 | 26 | 22.6 | 21 | 17.1 |
| Nov-17 | 70,883 | 2.34 | 1,303 | 2,411 | 23,287 | 1.84 | .835 | .130 | 28 | 23.5 | 22 | 17.0 |
| Oct-17 | 66,371 | 2.35 | 1,257 | 2,326 | 27,750 | 1.59 | .850 | .125 | 25 | 22.7 | 16 | 13.0 |
| Sep-17 | 62,983 | 2.32 | 1,220 | 2,258 | 22,131 | 1.83 | .844 | .136 | 31 | 27.7 | 20 | 15.4 |
| Aug-17 | 58,824 | 2.30 | 1,192 | 2,212 | 24,029 | 1.68 | .838 | .106 | 22 | 18.2 | 15 | 11.4 |
| Jul-17 | 55,638 | 2.28 | 1,126 | 2,088 | 21,583 | 1.92 | .842 | .127 | 28 | 22.2 | 15 | 10.1 |
| Jun-17 | 52,438 | 2.31 | 1,068 | 1,972 | 16,278 | 1.92 | .833 | .124 | 22 | 16.7 | 19 | 13.9 |
| May-17 | 48,009 | 2.35 | 1,014 | 1,870 | 23,046 | 1.84 | .837 | .124 | 27 | 21.8 | 13 | 8.8 |
| Apr-17 | 43,855 | 2.37 | 966 | 1,778 | 21,420 | 1.90 | .835 | .114 | 27 | 21.3 | 14 | 9.4 |

Fig 6 shows the scaled density distributions of the estimated risk values for the September 2022 validation cohort, split by those with and without the outcome. To facilitate visual examination, the x-axis is cut slightly above 25%. Note: only ~0.16% of patients from this cohort were assigned risk values >25%–the full range, median, mean, and IQR are provided in Table 3.

Fig 6 and Table 3 indicate positively skewed distributions for both groups, illustrating that patients are typically assigned low risk values–this is expected due to the low incidence rate of the PoC endpoint. However, when compared to patients who do not experience the endpoint, those who do have a more dispersed distribution away from the lowest end of the risk range. This demonstrates how the model distinguishes between patients at different classification cut points: for a given risk value, a larger proportion of patients with the endpoint (relative to those without) are positioned within the right tail. For instance, 38.4% and 4.4% of patients with and without the endpoint, respectively, have an estimated risk ≥5%, while 58.7% and 10.2% of patients with and without the endpoint, respectively, have an estimated risk ≥2.5%. Consequently, the 5% threshold for high vs low risk classification is linked to a specificity of 95.6% and a sensitivity of 38.4%, while the 2.5% threshold corresponds to a specificity of 89.8% and a sensitivity of 58.7%.

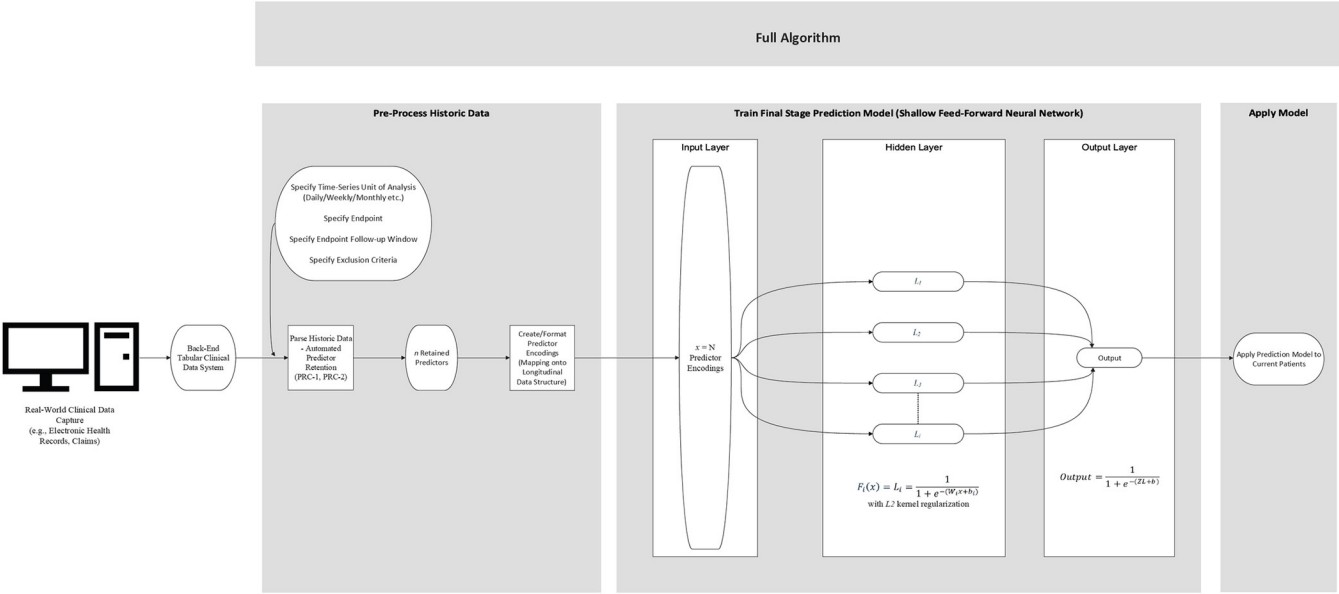

**Fig 2. Full Algorithm Visualization: Pre-Processing, Model Training, and Model Application Sections.**

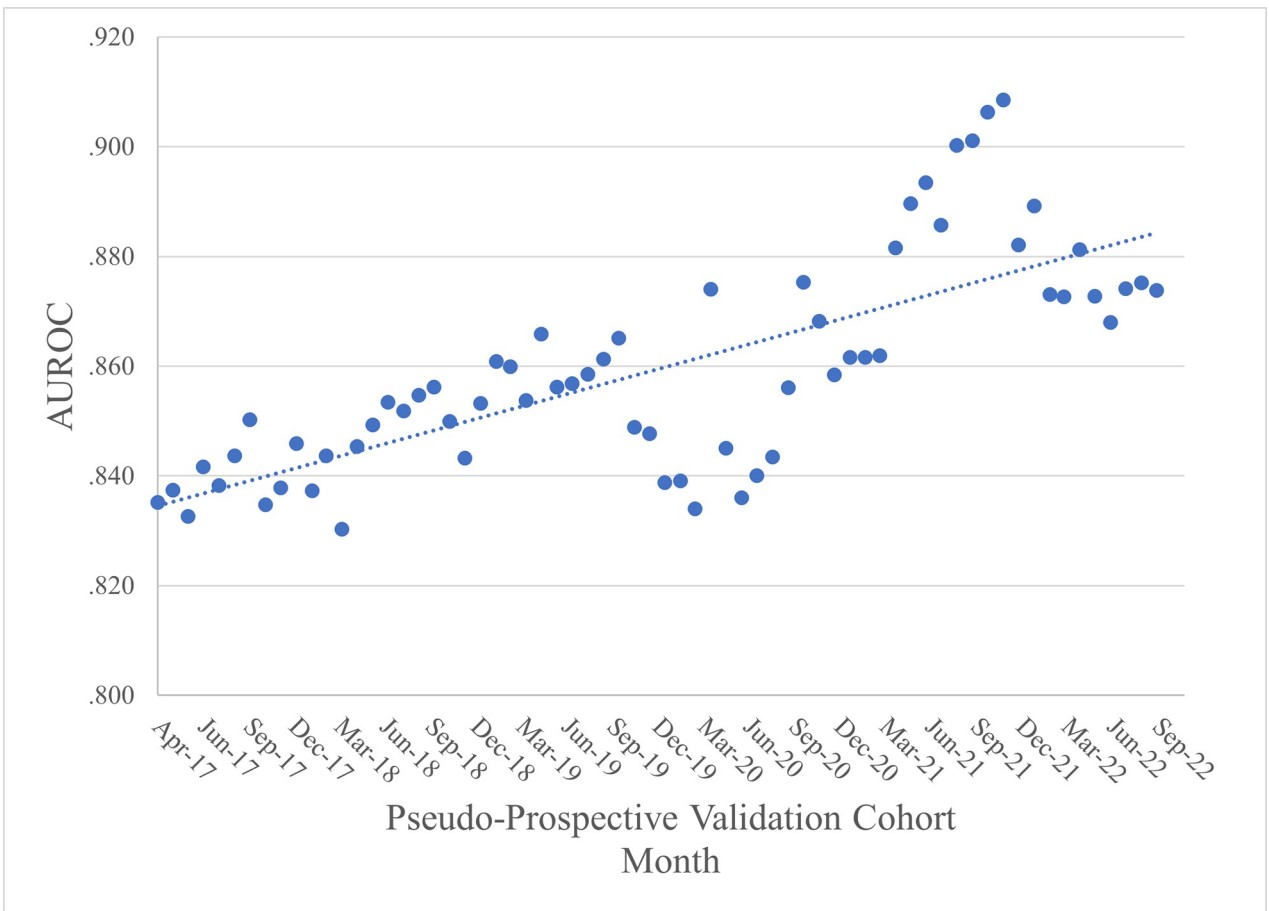

**Fig 3. Predicting 12-Month Initial Stroke/MI: AUROC Across Validation Cohorts.**

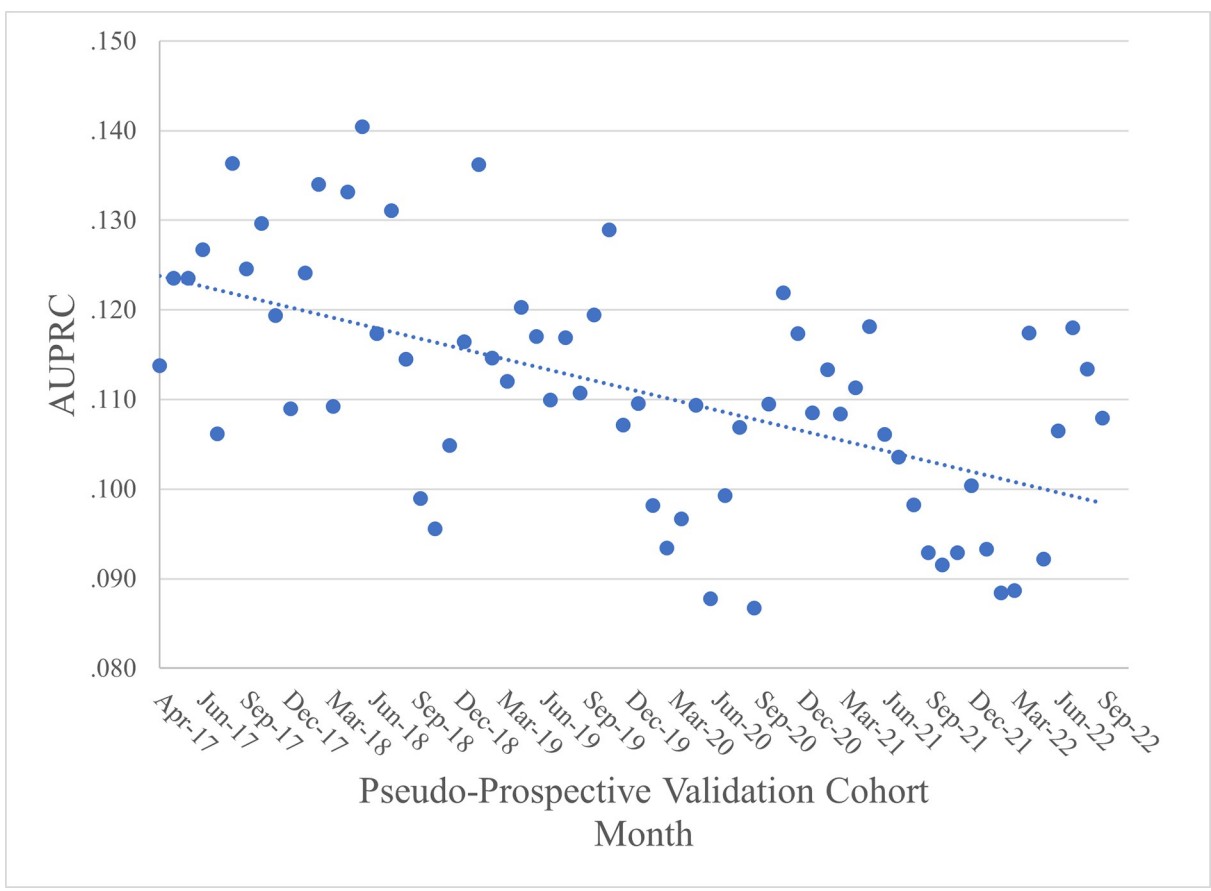

**Fig 4. Predicting 12-Month Initial Stroke/MI: AUPRC Across Validation Cohorts.**

Lastly, Table 2 indicates that the automated predictor selection strategy tends to incorporate a greater number of retained predictors over time. This trend reflects the impact of new patient data: as additional data are processed through consecutive runs of the full algorithm, BPs that previously did not meet the two PRCs become integrated into the models when empirical evidence suggests their inclusion may benefit the prediction. Although this trend generally increases over time, it is possible for a BP to be removed from the model if new data suggest that it no longer contributes to the prediction. This too can enhance model performance by eliminating noise from the predictor-space prior to training.

## Discussion

### Example PoC

AUROC values are good to excellent and improve over time across successive runs, while AUPRC values are low and decline over time. However, it is crucial to recognize that AUROC and AUPRC metrics summarize performance across the entire estimated risk spectrum, and these low AUPRC values in particular reflect the rarity of the endpoint within a universal, endpoint-naïve hospital population. In practice, resource constraints, such as clinic capacity, impose strict limitations on the number of patients who can be manually screened based on risk estimates from systems processing large batches of patients simultaneously. Therefore, when interpreting risk estimates for the example PoC, the rank-ordered strategy discussed above offers more utility compared to the threshold-based classification strategy. Shifting

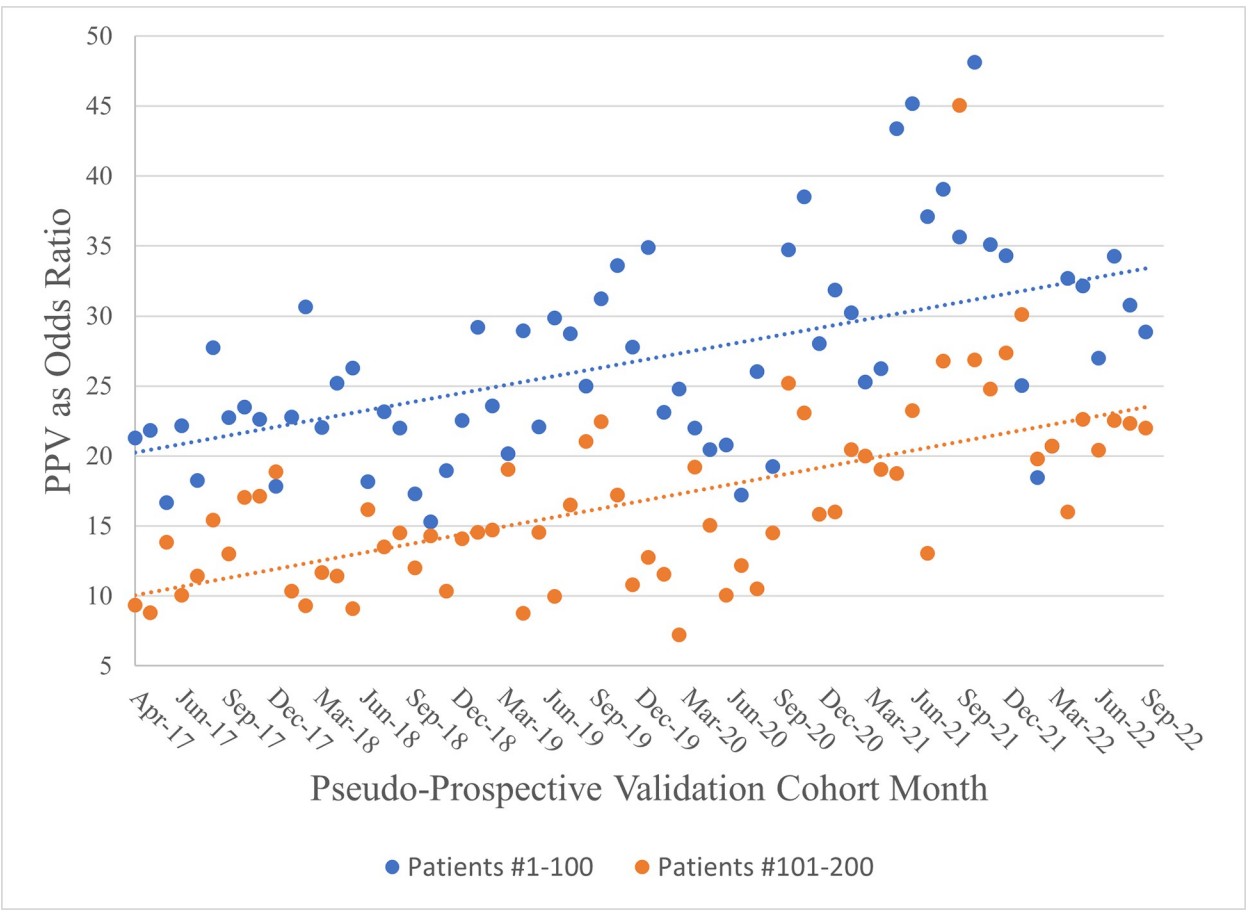

**Fig 5. Predicting 12-Month Initial Stroke/MI: PPV as Odds Ratio Across Validation Cohorts.**

focus to metrics describing model performance for those estimated to be at highest risk is therefore appropriate.

Results reveal that for patients ranked 1–100 and 101–200, PPVs range from 17% to 33% and 8% to 24% respectively, with average values of 25% and 17.2% respectively. To provide context for PPV values, we provide associated ORs, which standardize PPV by incorporating the endpoint's incidence rate in each validation cohort. For patients ranked 1–100 and 101–200, PPV OR ranges are 15.3 to 48.1 and 7.2 to 45.0 respectively. Notably, an upward/improving trend exists in ORs indicating that precision effect size tends to increase for the highest risk patients over time as the algorithm integrates/learns from newly collected patient data.

To further contextualize this level of precision, OR values describing the increased odds of breast and ovarian cancer associated with pathogenic genetic mutations BRCA1 and BRCA2 were calculated based on published incidence rates. For breast cancer, BRCA1 and BRCA2 OR values are approximately 11.6 and 8.9 respectively, while for ovarian cancer, BRCA1 and BRCA2 OR values are approximately 58.4 and 13.4 respectively [28–30]. When compared to OR values from the example PoC, being ranked in the top 200 highest risk patients in a given month provides predictive information comparably informative (if not more informative) to being found to carry pathogenic mutations in BRCA1 and BRCA2 genes. As the PoC endpoint is neither breast nor ovarian cancer, these data do not suggest that our system is more informative than BRCA1 and BRCA2 for these malignancies specifically. However, this comparison

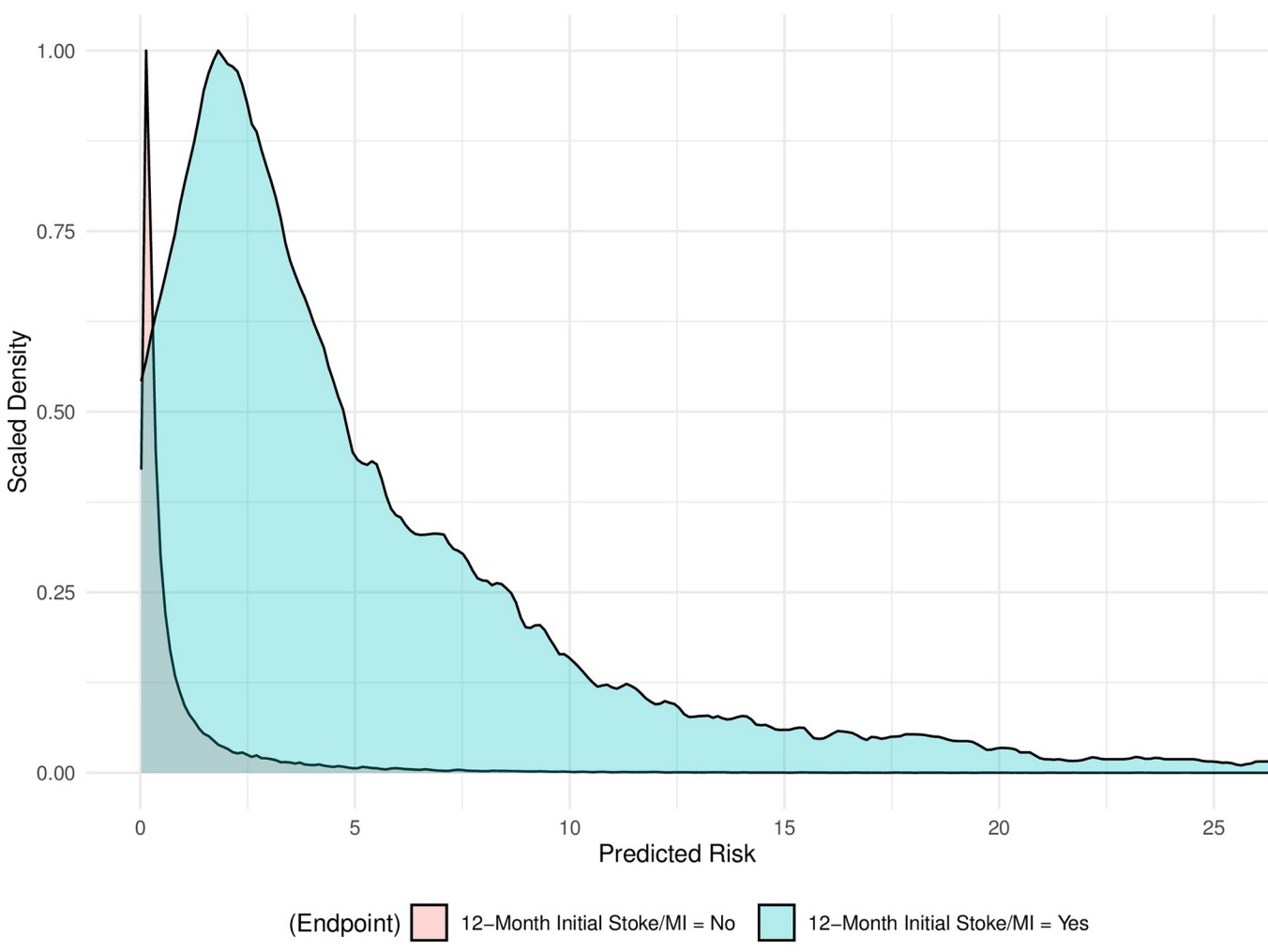

**Fig 6. Predicted Risk Scaled Density Distributions. Pseudo-Prospective Validation Cohort = September 2022.**

faithfully underscores the relative magnitude of the predictive information produced by the full algorithm when applied our hospital's EHR to predict 12-month risk of initial stroke/MI.

Overall, these results strongly indicate that across successive monthly runs, the full algorithm consistently produces high-performing predictions for patients with the highest estimated risk values, and even improves this performance over time as it incorporates/learns from newly collected patient data to refine its prediction mechanism.

### General advantages of this strategy/algorithm

Developing disease risk calculators as described in this manuscript offers two key advantages: (1) resulting predictions reflect health dynamics captured from data representing both

**Table 3. Predicted Risk Distributional Descriptives. Pseudo-Prospective Validation Cohort = September 2022.**

| 12-Month Initial Stroke/MI | Min | 1st Quartile | Median | Mean | 3rd Quartile | Max |
|---|---|---|---|---|---|---|
| No | 0.020 | 0.144 | 0.318 | 1.059 | 0.891 | 56.303 |
| Yes | 0.071 | 1.444 | 3.293 | 6.017 | 7.666 | 52.139 |

historical and contemporary local patients, and (2) predictions are functions of both well-known and novel risk/protective factors.

Another practical benefit of this approach is that it is feasible to undertake locally due to its reliance on tabular data. While an appropriately powerful computer system is required for modelling tabular data in this manner, it does not demand the high-end computer resources essential for other clinical-AI applications that process large quantities of unstructured data (e.g., language models, computer vision). Additionally, because our approach does not rely on transformer or long short-term memory NN architectures, this further reduces the development, hardware, and training-time requirements, all of which are of practical importance for local institutions hoping to perform such work.

If requisite data are available locally, our experience suggests that a small expert team can perform development. For the example PoC, only a single workstation was required, coding was performed by a single biostatistician, and consulting related to data context and clinical questions was provided by a team of four healthcare workers.

However, our project has its limitations.

## A targeted subpopulation

Implementing more targeted inclusion/exclusion criteria may further enhance model performance by restricting the applicable population to a pre-selected group of elevated-risk individuals based on established clinical theory or clinical-care context. This refinement can increase the incidence rate of the endpoint in the data splits, subsequently improving model performance related to AUPRC. However, the trade-off is that a targeted strategy will also reduce the system's ability to serve as a universal screening tool. Such a targeted system may overlook high-risk patients simply because they did not meet the pre-defined inclusion/exclusion criteria that trigger risk estimation.

## Requisite data

To enable this type of risk prediction, a baseline level or quantity of data must be available within the local data system. Predictions may not be successful or appropriate for endpoints that occur in an exceedingly small number of patients within the data system. Additionally, because much of the design of the full algorithm assumes high dimensionality in the predictor space, it may not be appropriate to apply this strategy to a structured data system that has low dimensionality compared to a full EHR or claims database (e.g., those resulting from registries). Ultimately, we recommend that a pseudo-prospective validation study is always performed to assess the algorithm's ability to formulate accurate predictions for a given data system.

## Manual and prospective validation studies

Prior to operationalizing the full algorithm, a manual validation study should also be conducted. Clinical staff should examine records from patients identified as high risk by the algorithm to assess the reasonableness of the prediction. This can be undertaken using retrospective data and can complement the pseudo-prospective validation strategy illustrated in this manuscript. Such work will help illuminate the true clinical utility of the algorithm, provide an opportunity for clinicians to provide feedback that can be incorporated into the algorithm's specification, and familiarize clinicians with the system before its operationalization.

Additionally, a truly prospective validation study may also be useful. Here, it might be uncovered that even though the algorithm identifies high-risk patients in a manner that is not always reasonable to clinicians, the identified patients nonetheless experience the endpoint at

expected rates as predicted by the model. Such findings would contribute to the confidence placed in the prediction system and may even begin to modify clinical understanding of risk in the local population.

### Bivariate automated predictor retention criteria

The two predictor retention criteria we utilize only examine the relationship between a potential predictor variable and the endpoint in a bivariate context. When variable $X$ is predictive of the endpoint, but only in a multi-variate context, PRC-1 and PRC-2 will fail to retain $X$. This omission represents a lost opportunity to incorporate informative data into the prediction. Expanding future algorithm specifications that integrate multi-variate predictor retention criteria may improve model performance as the resulting predictor space will more fully incorporate the relevant predictive information contained within the underlying data system.

### Numeric data

For reasons discussed above we opted not to integrate numeric predictors capturing lab results, flowsheet measurements, and medication dosage into our algorithm, despite their availability in our hospital's EHR. Future research should explore whether incorporating these numeric data enhances prediction sufficiently to justify the additional computational resources required for their inclusion.

For the numeric variables that were utilized in the current algorithm specification, only the last chronological value within a patient-month was retained. An alternative strategy that incorporates the distributional characteristics of multiple measurements (e.g., mean, min, max) could more comprehensively capture these numeric data and potentially enhance prediction further.

### Longitudinal NN specification

The NN architecture we used did not link multiple rows from the same patient to each other. Alternative specifications that explicitly account for the clustering of rows within patients (e.g., grand-mean-centered fixed effects, random effects, transformers, LSTM) may enhance model performance by better addressing within-patient correlations over time. However, implementing these methods will necessitate the use of higher computer resources during model training. Future research should evaluate whether such strategies sufficiently enhance prediction to justify the increased resource demands.

### A more informative outcome

The binary outcome scheme we employed does not explicitly differentiate between the immediacy of risk. Additionally, this scheme necessitates strict adherence to follow-up windows before coding the outcome, resulting in a delayed ability of the full algorithm to utilize contemporary data when constructing the analytic dataset. Modifying the outcome specification to enable direct risk estimation for different time intervals (e.g., 1-to-3 months vs. 4-to-12 months) may enhance the system's utility. Moreover, utilizing outcome variable and model specifications resembling survival analysis methods could also improve utility by incorporating more information into the outcome variable and allowing flexibility in the required length of the follow-up window.

### Errors in the data

In general, accurate variable coding depends on complete and proper data entry in the real-world data system. Besides those due to human error, omissions in the data can occur when

patients receive care outside of the facility/system captured by the data source and these data are not later entered into the data source. Access to cross-facility EHR and/or insurance data could improve the completeness of captured clinical histories, potentially enhancing the performance of the resulting prediction models.

Specific types of data errors pose varying threats to the full algorithm's ability to formulate a prediction model (which is trained on historic data) that generalizes to current/unseen validation data. For instance, random errors in the endpoint or predictor-space can affect the model's estimation of a patient's true risk of the endpoint but won't systematically bias the model's ability to rank-order patients by relative risk. This is because random errors are by definition distributed randomly in the data, thereby eliminating their ability to influence model coefficients for those with and without the endpoint in a systematically variable manner. Using a rank-ordered screening strategy therefore provides protection against random errors in the data, and thus, is recommended when possible.

Systematic errors in the endpoint, on the other hand, pose a threat to model validity because they may result in the model providing risk estimates that reflect an outcome whose definition varies from that of the desired endpoint. For example, if hyperlipidemia is systematically coded into patient records simply to assist with insurance claims approvals, and this diagnosis is set as the endpoint of interest, then the final-stage prediction model will predict an endpoint that reflects this coding practice. We therefore recommend only targeting endpoints if local clinical staff are confident in the overall accuracy of their coding. Focusing on severe/acute diagnostic codes (as we did in the example PoC) might be a strategy here, as these codes are less likely (relative to chronic conditions) to be entered into patient records to simply satisfy claims requirements. Additionally, defining the outcome using a set of variables whose combined presence indicates the endpoint is another strategy to help in this regard.

Systematic errors in the predictor-space will not bias model prediction if such errors are consistent in both the historic/analytic and current/unseen validation data. In fact, though such errors do not reflect clinical reality, they may actually aid the prediction if their dynamics follows consistent patterns across both historic and current datasets.

Alternatively, when systematic errors in the predictor-space are not consistent across time, they pose a threat to model generalizability because the prediction mechanism (which is based on historic/analytic data) will be misaligned with current/unseen validation data. In response to this threat, we employed safeguards within the NN architecture and estimation that limits the model's ability to capture precise patterns in the analytic dataset.

Additionally, as stated above, we hypothesized that the full algorithm would be robust to data errors and noise when applied to real-world clinical tabular data systems–because such systems are by their nature highly correlated and high dimensional, it is plausible that the prediction signal will be able to successfully traverse a predictor-space containing errors/noise because it will have multiple complementary pathways to travel. Though we did not formally test this hypothesis, by applying the full algorithm to a live uncurated EHR, we were able to empirically show that this hypothesis seems plausible on the surface. Moreover, given the promising results found here, our work suggests that a-priori data curation may not be an absolute requirement when developing predictive algorithms in this manner. Of course, formal examination of this hypothesis is strongly recommended as a direction for future research, as this will help clarify the potential of such algorithms as a subset of the clinical-AI landscape.

## Supporting information

**S1 Appendix. Retained Predictors—Validation Cohort = September 2022.**
(XLSX)

## Author Contributions

**Conceptualization:** Terrence J. Lee-St. John, Bartlomiej Piechowski-Jozwiak.

**Data curation:** Terrence J. Lee-St. John.

**Formal analysis:** Terrence J. Lee-St. John.

**Investigation:** Terrence J. Lee-St. John, Emna Abidi, Wasim El Nekidy, Bartlomiej Piechowski-Jozwiak.

**Methodology:** Terrence J. Lee-St. John, Oshin Kanwar, Wasim El Nekidy, Bartlomiej Piechowski-Jozwiak.

**Project administration:** Terrence J. Lee-St. John.

**Supervision:** Terrence J. Lee-St. John.

**Validation:** Terrence J. Lee-St. John.

**Visualization:** Terrence J. Lee-St. John.

**Writing – original draft:** Terrence J. Lee-St. John.

**Writing – review & editing:** Terrence J. Lee-St. John, Oshin Kanwar, Emna Abidi, Wasim El Nekidy, Bartlomiej Piechowski-Jozwiak.

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
