## [Decision Letter · Decision Letter 0]

18 Jun 2024

PDIG-D-24-00181

Towards Artificial Intelligence-Based Disease Prediction Algorithms

that Learn from Real-World Clinical Tabular Data Systems

PLOS Digital Health

Dear Dr. Lee St John,

Thank you for submitting your manuscript to PLOS Digital Health. After careful consideration, we feel that it has merit but does not fully meet PLOS Digital Health's publication criteria as it currently stands. Therefore, we invite you to submit a revised version of the manuscript that addresses the points raised during the review process.

Please submit your revised manuscript within 60 days Aug 17 2024 11:59PM. If you will need more time than this to complete your revisions, please reply to this message or contact the journal office at digitalhealth@plos.org. Please include the following items when submitting your revised manuscript:

We look forward to receiving your revised manuscript.

Kind regards,

Hualou Liang

Academic Editor

PLOS Digital Health

Journal Requirements:

1. We ask that a manuscript source file is provided at Revision. Please upload your manuscript file as a .doc, .docx, .rtf or .tex.

2. Please provide separate figure files in .tif or .eps format only and remove any figures embedded in your manuscript file. Please also ensure that all files are under our size limit of 10MB.

3. In the online submission form, you indicated that "EHR data utilized during this project are not publicly available due to HIPAA and local privacy laws. Deidentified extracts are available from the corresponding author upon reasonable request but will require local Department of Health approval in alignment with local privacy laws". 

3. Uploaded as supplementary information.

Additional Editor Comments (if provided):

Reviewers' comments:

Reviewer's Responses to Questions

**Comments to the Author**

1. Does this manuscript meet PLOS Digital Health’s publication criteria? Is the manuscript technically sound, and do the data support the conclusions? The manuscript must describe methodologically and ethically rigorous research with conclusions that are appropriately drawn based on the data presented.

Reviewer #1: Yes

Reviewer #2: Yes

Reviewer #3: Yes

2. Has the statistical analysis been performed appropriately and rigorously?

Reviewer #1: Yes

Reviewer #2: Yes

Reviewer #3: Yes

3. Have the authors made all data underlying the findings in their manuscript fully available (please refer to the Data Availability Statement at the start of the manuscript PDF file)?

Reviewer #1: No

Reviewer #2: No

Reviewer #3: Yes

4. Is the manuscript presented in an intelligible fashion and written in standard English?

Reviewer #1: Yes

Reviewer #2: Yes

Reviewer #3: No

5. Review Comments to the Author

Reviewer #1: This paper addressed a common problem faced by physicians and presented a classic case of how AI may help clinical practice. However, the technical depth needs to be strengthened. There should be more in-depth discussion on the structure/type of the neural network used, what was the parameter set up, what was the considerations, etc. Otherwise it'll be simply another case of using an off the shelf model on some data. In addition, the clinical relevance of the result should be included.

Reviewer #2: The manuscript presents a proof-of-concept for an algorithm that could revolutionize disease risk estimation using real-world clinical data. By ingeniously combining classic statistical methods with cutting-edge artificial intelligence techniques, the researchers have developed a versatile algorithm capable of autonomously navigating through complex clinical data systems like electronic health records (EHR) or claims databases. What sets this algorithm apart is its self-adaptive nature, constantly evolving to reflect changes within the data system, thus ensuring the accuracy and relevance of risk estimates over time.

The comprehensive approach of the algorithm, which considers a multitude of factors including encounters, diagnoses, procedures, medications, labs, and demographics, without the need for extensive data curation, is a remarkable departure from traditional static prediction models. The authors' rigorous validation process, spanning a significant timeframe and involving a large patient cohort, demonstrates the algorithm's robustness and effectiveness in estimating the risk of critical events like stroke or myocardial infarction.

The reported AUROC values and odds ratios provide compelling evidence of the algorithm's performance and its potential to significantly enhance clinical decision-making. Moreover, the observed improving trend over time underscores the algorithm's capability to continuously learn and adapt, promising even greater accuracy and reliability in the future.

In summary, this manuscript represents a major advancement in the field of predictive analytics in healthcare. The algorithm's ability to harness the full spectrum of information within real-world clinical data systems, coupled with its adaptive nature and impressive performance metrics, paves the way for the development of highly effective disease risk calculators that can truly revolutionize patient care.

Reviewer #3: In my opinion, although the faced problem is interesting, the work needs further improvements in order to reach a version to publish and that could effictively used by the sceintific community.

First, It's unclear how the integration of social behavior factors with environmental variables contributes to more accurate predictions. I suggest proofreading since the paper is affected by several typos and grammatical errors. Furthermore, the lack of a precise and detailed discussion about the direction of the article, and specifically a clear explanation of motivation, scenario and overall contribution make the paper not easier to follow and understand.

Second, the proposed method appears overly complex with the integration of multiple neural network architectures like LSTM, gate recurrent unit, BiLSTM, and attention mechanisms. This complexity might hinder the method's practicality and interpretability. This could help the reader to understand the novelty introduced by the paper. Actually, they not evident the advantages of the proposed method in comparison with the existing works.

While the abstract mentions experimental results, it doesn't provide insights into how the proposed method performs in real-world scenarios. Without validation against real-world data, the practical relevance and generalizability of the method remain questionable.

Due to the reasons given above, this paper should be rejected.

6. PLOS authors have the option to publish the peer review history of their article (what does this mean?). If published, this will include your full peer review and any attached files.

**Do you want your identity to be public for this peer review?** For information about this choice, including consent withdrawal, please see our Privacy Policy.

Reviewer #1: No

Reviewer #2: No

Reviewer #3: Yes: Surjeet Dalal

---

## [Decision Letter · Decision Letter 1]

21 Jul 2024

Towards Artificial Intelligence-Based Disease Prediction Algorithms

that Learn from Real-World Clinical Tabular Data Systems

PDIG-D-24-00181R1

Dear Dr. Lee St John,

We are pleased to inform you that your manuscript 'Towards Artificial Intelligence-Based Disease Prediction Algorithms

that Learn from Real-World Clinical Tabular Data Systems' has been provisionally accepted for publication in PLOS Digital Health.

Best regards,

Hualou Liang

Academic Editor

PLOS Digital Health

Reviewer Comments (if any, and for reference):

Reviewer's Responses to Questions

**Comments to the Author**

1. If the authors have adequately addressed your comments raised in a previous round of review and you feel that this manuscript is now acceptable for publication, you may indicate that here to bypass the “Comments to the Author” section, enter your conflict of interest statement in the “Confidential to Editor” section, and submit your "Accept" recommendation.

Reviewer #1: All comments have been addressed

2. Does this manuscript meet PLOS Digital Health’s publication criteria? Is the manuscript technically sound, and do the data support the conclusions? The manuscript must describe methodologically and ethically rigorous research with conclusions that are appropriately drawn based on the data presented.

Reviewer #1: Yes

3. Has the statistical analysis been performed appropriately and rigorously?

Reviewer #1: Yes

4. Have the authors made all data underlying the findings in their manuscript fully available (please refer to the Data Availability Statement at the start of the manuscript PDF file)?

Reviewer #1: No

5. Is the manuscript presented in an intelligible fashion and written in standard English?

Reviewer #1: Yes

6. Review Comments to the Author

Reviewer #1: Comments have been addressed by authors.

7. PLOS authors have the option to publish the peer review history of their article (what does this mean?). If published, this will include your full peer review and any attached files.

**Do you want your identity to be public for this peer review?** For information about this choice, including consent withdrawal, please see our Privacy Policy.

Reviewer #1: No
